# Enhancing Leaf Area Index Estimation with MODIS BRDF Data by Optimizing Directional Observations and Integrating PROSAIL and Ross–Li Models

Hu Zhang [1], Xiaoning Zhang [2], Lei Cui [1], Yadong Dong [3], Yan Liu [3,*], Qianrui Xi [1], Hongtao Cao [1], Lei Chen [1] and Yi Lian [1]

1. School of Geographic and Environmental Sciences, Tianjin Normal University, Tianjin 300387, China; huzhang@tjnu.edu.cn (H.Z.); cuil@mail.bnu.edu.cn (L.C.); 2210080037@stu.tjnu.edu.cn (Q.X.); caoht@tjnu.edu.cn (H.C.); lchen@tjnu.edu.cn (L.C.); lianyi@tjnu.edu.cn (Y.L.)
2. School of Mechatronical Engineering, Beijing Institute of Technology, Beijing 100081, China; xnzhang@bit.edu.cn
3. Aerospace Information Research Institute, Chinese Academy of Sciences, Beijing 100094, China; dongyd@aircas.ac.cn
* Correspondence: liuyan@aircas.ac.cn

**Abstract:** The Leaf Area Index (LAI) is a crucial vegetation parameter for climate and ecological models. Reflectance anisotropy contains valuable supplementary information for the retrieval of properties of an observed target surface. Previous studies have utilized multi-angular reflectance data and physically based Bidirectional Reflectance Distribution Function (BRDF) models with detailed vegetation structure descriptions for LAI estimation. However, the optimal selection of viewing angles for improved inversion results has received limited attention. By optimizing directional observations and integrating the PROSAIL and Ross–Li models, this study aims to enhance LAI estimation from MODIS BRDF data. A dataset of 20,000 vegetation parameter combinations was utilized to identify the directions in which the PROSAIL model exhibits higher sensitivity to LAI changes and better consistency with the Ross–Li BRDF models. The results reveal significant variations in the sensitivity of the PROSAIL model to LAI changes and its consistency with the Ross–Li model over the viewing hemisphere. In the red band, directions with high sensitivity to LAI changes and strong model consistency are mainly found at smaller solar and viewing zenith angles. In the near-infrared band, these directions are distributed at positions with larger solar and viewing zenith angles. Validation using field measurements and LAI maps demonstrates that the proposed method achieves comparable accuracy to an algorithm utilizing 397 viewing angles by utilizing reflectance data from only 30 directions. Moreover, there is a significant improvement in computational efficiency. The accuracy of LAI estimation obtained from simulated multi-angle data is relatively high for LAI values below 3.5 when compared with the MODIS LAI product from two tiles. Additionally, there is also a slight improvement in the results when the LAI exceeds 4.5. Overall, our results highlight the potential of utilizing multi-angular reflectance in specific directions for vegetation parameter inversion, showcasing the promise of this method for large-scale LAI estimation.

**Keywords:** leaf area index (LAI); kernel-driven Ross–Li model; PROSAIL model; MODIS BRDF

## 1. Introduction

The leaf area index (LAI), occupying a crucial role in the vegetation canopy [1], is defined as the hemi-surface area of all leaves or needles in the vegetation canopy divided by the horizontal ground surface area [2,3]. LAI serves as a key indicator of vegetation health, productivity, and overall ecosystem functioning [4]. It provides essential information about the size and distribution of the leaf canopy, relating to various processes such as photosynthesis [5], evaporation and transpiration [6,7], and carbon assimilation [8]. Accurately estimating the LAI is of utmost importance as it has been employed in estimating

biophysical parameters of vegetation [9,10] and in monitoring vegetation growth [11], as well as in the study of global climate change [12], ecosystem productivity, biogeochemistry, hydrology, and ecology [13]. LAI also plays a crucial role in modeling the exchanges of mass, energy, and momentum between the biosphere and the atmosphere [14,15]. Therefore, ensuring the accuracy of LAI estimation is pivotal for effective ecological research, environmental monitoring, and sustainable land management strategies.

The measurement of the LAI can be achieved through various approaches [16]. Direct methods entail partially or completely defoliating the canopy to assess the total leaf area of plants or trees, which involves destructiveness and a time-consuming process [17,18]. Indirect approaches employ mathematical algorithms to characterize the passage of light through the canopy, utilizing Beer's Law for estimating the total leaf area [16,19]. In recent years, technological advancements have revolutionized the way we measure leaf area. Surface reflectance data captured by moderate-resolution sensors have been employed in the generation of several global LAI products [20–24], and these products have been extensively validated using field measurements and upscaled high-resolution LAI reference maps [25–29]. These technologies offer the potential to enhance our understanding of vegetation dynamics over vast geographical areas, providing valuable information for global climate change studies. However, accurate measurements face challenges posed by factors like cloud cover, satellite orbit constraints, and the requirement for precise multi-angle data [30].

Ground-based experiments have demonstrated that utilizing abundant multi-angle information can significantly improve LAI estimation accuracy [31]. This improvement is attributed to the fact that multi-angle data provide a wealth of additional information about vegetation parameters. Satellite-observed reflectance at multiple viewing angles, along with physical BRDF models that provide detailed descriptions of vegetation structure, are often directly utilized for the retrieval of vegetation structure parameters [32]. Physical BRDF models, based on physical laws, describe how radiation interacts with vegetation canopies. These models enable the calculation of reflectance under different viewing and illumination conditions, taking into account a wide range of leaf and canopy structure parameters. By employing different inversion techniques, physical BRDF models can estimate these vegetation parameters using reflectance data as input [33]. For example, using the PROSAIL model can simulate the interaction of light with vegetation at different scales, from individual leaves to entire canopies, and provides outputs such as leaf reflectance, transmittance, and absorptance, as well as canopy reflectance [34,35]. By employing the PROSAIL model in inverse mode, it is possible to obtain biophysical parameters such as LAI, chlorophyll content, and vegetation water content from canopy reflectance [36].

Although multi-angular measurements provide enhanced information regarding the structural properties of vegetation [37–40], it is important to acknowledge that the capability of satellites to acquire multi-angle data is constrained by the characteristics of their orbits as well as by the presence of clouds and shadows [41], leading to a loss of anisotropic information. The number of angle samples in remote sensing observations often fails to meet the sample quantity requirements of the physical BRDF model inversion. Additionally, reflectance measurements with similar geometric configurations tend to exhibit significant autocorrelation, while random noise in these measurements is closely associated with vegetation structure [42]. Moreover, during the inverse process, iterative optimization techniques may encounter the challenge of converging to a local minimum instead of the global minimum [31], which can result in potentially inaccurate outcomes. All the above-mentioned factors may have an impact on the accuracy of LAI inversion.

By accumulating observations over time from satellite sensors with a wide field of view, such as those onboard the Moderate Resolution Imaging Spectroradiometer (MODIS) [43] and the Multi-angle Imaging SpectroRadiometer (MISR) [44], along with Ross–Li models, BRDF products have been successfully generated at both global and regional scales. The surface BRDF product provides a means to characterize the directional properties of the underlying surface reflection across the entire observation hemisphere [45,46]. Coupling

different models is a commonly employed strategy to utilize their individual strengths and enhance the accuracy of modeling surface reflection and the estimation of vegetation parameters. A model-to-model approach has been proposed to avoid most of the challenges associated with inverting LAI, which by using multi-angular reflectance can be resolved [47]. In this kind of approach, the multi-angular measurements are used to fit a simplified BRDF model that requires only a few parameters to describe the surface reflectance anisotropy. For example, by fitting the Ross–Li BRDF model to the multi-angular reflectance data, the model provides estimates of the BRDF parameters, which can quantitatively describe how surface reflectance varies with different viewing and illumination angles [42,48]. Parameters of the model are then estimated to facilitate the inversion process. The BRDF information reconstructed from these satellite observations has been successfully applied in numerous parameter estimations, such as albedo, clumping index and canopy height [43,49–53], laying the foundation for improving LAI satellite inversion algorithms. Although the model-to-model approach provides a solution for using multi-angle observation to inverse LAI, it may introduce additional sources of error or uncertainty due to the additional modeling step.

Zhang et al. [30] coupled the PROSAIL model with the hotspot-corrected RossThick–LiSparseReciprocal (RTLSR_C) BRDF model [54,55] to inverse LAI. They used the PROSAIL model to simulate reflectance for 397 viewing directions and established a lookup table based on a dataset of 20,000 average distributed vegetation parameter combinations, and then utilized MODIS BRDF products to simulate directional reflectance, and subsequently inverted the data to obtain estimates of LAI. There is a disadvantage to this kind of method; that is, they did not search for the optimal number and positions of viewing angles that are needed to obtain a better inversion result. There are also studies suggesting that incorporating more observation directions introduces redundant information, thereby increasing the noise in the input data and leading to greater uncertainty in the retrieval of parameters [56]. Due to the varying ability of physical models to accurately reproduce anisotropy across different viewing directions, increasing the number of observations can also lead to a decrease in inversion accuracy [40]. Hence, it is crucial to enhance LAI estimation methods by incorporating reconstructed BRDF information and determining the optimal number and positions of viewing angles. This will improve the accuracy and efficiency of large-scale LAI estimation.

This study focuses on the collaborative use of the RTLSR_C and PROSAIL models, specifically examining the PROSAIL model's sensitivity to varying LAI at different observation positions. Our emphasis lies in ensuring consistency between the PROSAIL and RTLSR_C models across diverse observation points, with a primary goal of identifying optimal viewing angles for improved inversion results. The paper starts with an introduction highlighting the importance of LAI estimation. The methodology section details the coupling of models and the PROSAIL sensitivity analysis. Data utilization involves high-quality MODIS BRDF products and an improved look-up table for LAI inversion. Results from optimal viewing angles are presented, followed by validation against ground-based LAI measurements, high-resolution LAI maps, and MODIS LAI products.

## 2. Materials and Methods

The flowchart of this research is illustrated in Figure 1, which includes four main components. The first component focuses on analyzing the PROSAIL model's sensitivity to variations. The second component investigates the coherence between PROSAIL and RTLSR_C BRDF models under different observation geometries. The third component involves using the results obtained from the previous two steps to determine the optimal direction based on the analysis conducted with the 397 viewing directions. A new lookup table that captures the relationship between vegetation parameters and reflectance in the optimal observation direction using the PROSAIL model is created. The fourth component involves simulating the reflectance in the optimal observation direction based on MODIS BRDF products. By comparing the simulated reflectance and the corresponding reflectance

in the new lookup table, the minimum cost function is calculated to determine the LAI value. The results will be validated using ground-based LAI measurements and MODIS LAI products.

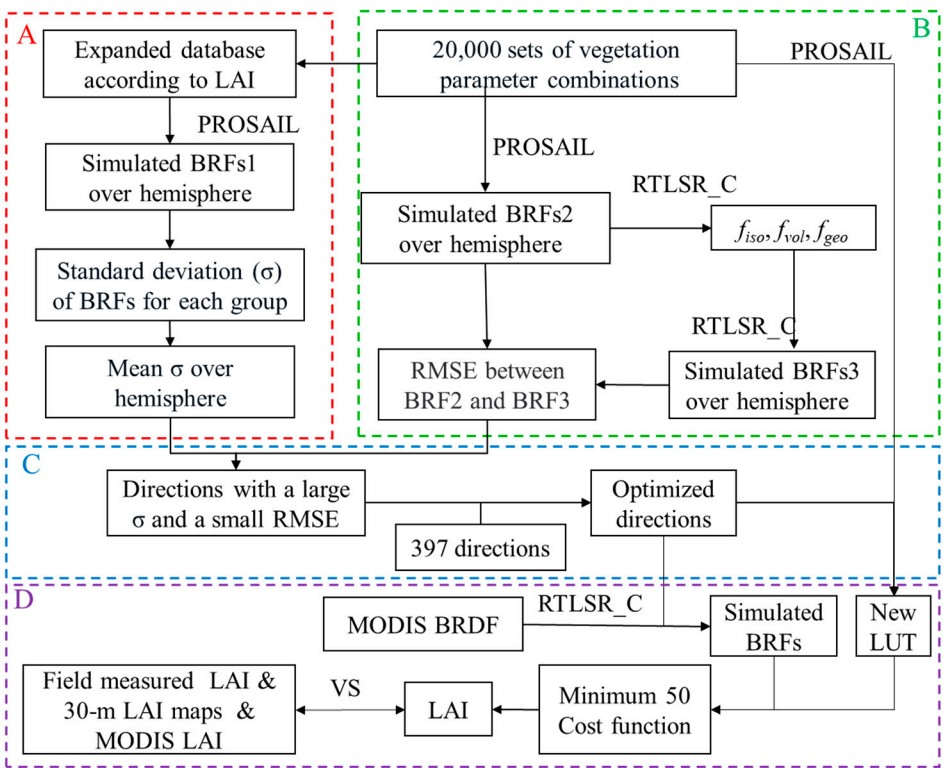

**Figure 1.** Flowchart of LAI estimation by linking the PROSAIL and Ross–Li BRDF models using the MODIS BRDF product. Part A is the PROSAIL model's sensitivity to LAI, part B is the coherence between PROSAIL and RTLSR_C BRDF models, part C is to determine the optimal direction, and part D is the inversion and validation of LAI based on MODIS BRDF.

### 2.1. PROSAIL Model for Multi-Angular Reflectance Simulations

The PROSAIL model, a reliable canopy radiative transfer model, was chosen for vegetation parameter inversion due to its simple input parameters and its ability to maintain a consistent BRDF with the Ross–Li model [57]. This study references the latest version of the 4SAIL canopy BRDF model, which incorporates the hotspot effect, along with the PROSPECT-5 model that provides a comprehensive description of leaf optical reflectance and transmittance (http://teledetection.ipgp.jussieu.fr/prosail/, accessed on 10 August 2021) [58–60]. The PROSAIL model has the ability to simulate canopy reflectance with a resolution of 1 nm for any sun-viewing geometry within the spectra range of 400–2500 nm.

Each parameter in the PROSAIL model is associated with a common value. This study utilizes a comprehensive simulation dataset of 20,000 vegetation parameter combinations, which was used in a previous study [30]. These combinations were generated using the Satellite periodic function through the uniform sampling of seven leaf and canopy parameters, i.e., leaf structure parameter (Ns), chlorophyll a and b content (Cab), equivalent water thickness (Cw), leaf mass per unit leaf area (Cm), leaf area index (LAI), average leaf angle (ALA), and soil coefficient (Psoil). Reasonable ranges have been set for all seven parameters, and the rest of the parameters have been set to the constant values. The specific information about these leaf and canopy parameters can be found in Zhang et al., 2021 [30].

By inputting the vegetation parameters into the PROSAIL model, the canopy reflectance spectra for the entire view hemisphere at different solar zenith angles can be simulated. The red and near-infrared (NIR) bands are preferred for retrieving most LAI products due to their heightened sensitivity [61]. Consequently, PROSAIL simulations

were executed at the central wavelengths within the two MODIS bands, specifically at 645 nm and 858 nm.

## 2.2. Kernel-Driven Ross–Li BRDF Model and MODIS BRDF

The kernel-driven Ross–Li model [42,48] has been widely adopted to reconstruct BRDF data in the sun-viewing hemisphere from limited observation. The RossThick–LiSparseReciprocal (RTLSR) model was selected as the operational algorithm for generating the MODIS BRDF parameter product [43] The general expression of the RTLSR model is provided in Equation (1). $R$ is the surface bidirectional reflectance in the wavelength $\lambda$; $f_{iso}$, $f_{vol}$, and $f_{geo}$ are the spectrally dependent BRDF model parameters; $K_{vol}$ and $K_{geo}$ are kernel functions of the volumetric [62,63] and geometric optical [48,64] scattering, respectively; $\Omega$ is a direction vector; and $i$ and $r$ represent the meanings of incident and reflected radiance.

$$R(\Omega_i, \Omega_r, \lambda) = f_{iso}(\lambda) + f_{vol}(\lambda)K_{vol}(\Omega_i, \Omega_r, \lambda) + f_{geo}(\lambda)K_{geo}(\Omega_i, \Omega_r, \lambda) \qquad (1)$$

Through the utilization of multi-angular reflectance data in the model, the three BRDF model parameters ($f_{iso}$, $f_{vol}$, and $f_{geo}$) can be derived by employing least-squares regression. Finally, by utilizing the extrapolation capability of the model, it is possible to simulate directional reflectance at arbitrary orientations [43,48].

The global 500 m BRDF parameter products provided by MODIS consist of spectrally dependent BRDF model parameters as well as quality data. In subsequent studies, only high-quality (full inversion, quality flag = 0/1) MODIS BRDF products [41,43] were used. In this study, a modified version of the Ross–Li model called RTLSR_C was employed [54,55]. The RTLSR_C model incorporates the corrected exponential hotspot function developed by Chen and Cihlar [65] into the volumetric and geometric-optical scattering kernels, resulting in an enhanced capability for simulating hotspots compared to the original RTLSR model. The optimal hotspot parameters (*C1* for height and *C2* for width) of the RTLSR_C model in various typical bands were determined through an extensive search using abundant hotspot data [54]. Due to the localized modification of reflectance near the hotspot, without affecting other directions, the MODIS BRDF parameters were directly applied to the hotspot-corrected kernels in order to simulate multi-angular reflectance. The hotspot parameters for MODIS at 645 nm (*C1* = 0.5, *C2* = 3.4°) and 858 nm (*C1* = 0.5, *C2* = 3.0°) were employed to simulate the directional reflectance using the MODIS BRDF parameters.

## 2.3. Determination of the Optimal Direction

When establishing a lookup table, it is important to choose an adequate number of directions to capture the anisotropic reflection characteristics of the Earth's surface. However, selecting too many directions can impact computational efficiency. It is necessary to evaluate the representativeness of reflectance in each selected direction. Based on the 397 sun-viewing geometry used in the previous study [30], this study focuses on identifying the directions in which the PROSAIL model exhibits a higher sensitivity to LAI changes and demonstrates better consistency with the RTLSR_C model. The range of solar zenith angles for the 397 observations is 0°–60° with a spacing of 15°. The range of viewing zenith angles is 0°–80° with a spacing of 10°. The range of relative azimuth angles is 0°–330° with a spacing of 30°. The reason for filtering the reflectance based on these directions is that the lookup table based on these directions yields a high accuracy of LAI, and this approach could avoid reflectance correlation in neighboring directions.

### 2.3.1. Sensitivity Analysis of the PROSAIL Model to Changes in LAI

To determine the most sensitive directions of the PROSAIL model to changes in LAI, the dataset of 20,000 sets of vegetation parameter combinations was expanded. For each vegetation parameter combination, the LAI value was varied from 0.5 to 10 with a step size of 0.5 while keeping the other parameters unchanged. Subsequently, the expanded vegetation parameters for each sample were sequentially inputted into the PROSAIL model, and the standard deviation ($\sigma$) (Equation (2)) of the simulated reflectance $\rho(\Omega_i, \Omega_r)$ for each

combination was calculated in any direction within the observed hemisphere. $n$ represents the sample size, and in this study its value is 20. $\overline{\rho}$ is the mean value of the 20 simulated sets of reflectance $\rho$. A larger value of σ indicates a higher sensitivity. To ensure a more representative outcome, the study finally employed the average value of σ obtained from 20,000 datasets.

$$\sigma(\Omega_i, \Omega_r) = \sqrt{\frac{\sum_{j=1}^{n}\left(\rho_j(\Omega_i, \Omega_r) - \overline{\rho(\Omega i, \Omega r)}\right)^2}{n-1}} \tag{2}$$

### 2.3.2. The Consistency between the Models

First, the 20,000 sets of vegetation parameters are sequentially input into the PROSAIL model to simulate the $\rho(\Omega_i, \Omega_r)$ in various directions. Then, the $\rho(\Omega_i, \Omega_r)$ is inputted into the RTLSR_C BRDF model to calculate the model parameters, and the reflectance $\rho_0(\Omega_i, \Omega_r)$ in any direction simulated by the kernel-driven model can be obtained. The root mean square error (RMSE) between the reflectance obtained by the two models (Equation (3)) is used to evaluate the consistency of the two models. The RMSE is computed for each direction using $\rho$ and $\rho_0$, with a value of $k$ set at 20,000. A lower RMSE value signifies greater consistency between the two models.

$$\text{RMSE}(\Omega_i, \Omega_r) = \sqrt{\frac{\sum_{j=1}^{k}\left(\rho_j(\Omega_i, \Omega_r) - \rho_{0,j}(\Omega_i, \Omega_r)\right)^2}{k-1}} \tag{3}$$

### 2.4. LAI Estimation from MODIS BRDF Data

Once the optimal directions were determined, in Section 2.3, the PROSAIL model was employed to simulate the reflectance in the red and NIR bands at these directions using 20,000 different combinations of vegetation parameters. As a result, a new lookup table was created to establish the relationship between vegetation parameters and multi-angle reflectance.

When calculating LAI using MODIS BRDF, the process begins by simulating the reflectance in the red and NIR bands at the specified directions using the MODIS BRDF product and the RTLSR_C model. Subsequently, the simulated reflectance values are compared with the reflectance values produced by the PROSAIL model in the new lookup table, and the $\text{RMSE}_c$ is computed over all available viewing angles and wavelengths. The calculation formula for $\text{RMSE}_c$ is the same as Equation (3), with the difference that $\rho$ represents the simulated reflectance in the red or NIR band at the selected direction when a specific set of vegetation parameter combinations is input into the PROSAIL model, while $\rho_0$ represents the corresponding simulated data from MODIS BRDF. Finally, the 50 records with the lowest $\text{RMSE}_c$ values are selected, and the corresponding LAI values are averaged to derive the retrieved LAI result [30].

To improve the computational efficiency, this study also introduced the linear relationships between the Area Leaf Angle Distribution (ALA) and the fraction of vegetation volume ($f_{vol}$) in the NIR band [30]. The formula is as follows:

$$ALA = 186.54 \times f_{NIR}^{vol} + 13.88 \left(ALA \in [10°, 85°], f_{NIR}^{vol} \in [0, 0.3813]\right) \tag{4}$$

During the actual computation process, the size of the lookup table is constrained by incorporating a variation of $\pm 3°$ around the empirically estimated Angular Leaf Alignment (ALA). To ensure that all data can participate in the calculation, when the $f_{vol}$ in the NIR band is not within the range of 0–0.3813, all 20,000 sets of data will be involved in the calculation.

### 2.5. Validation with LAI Measurements and MODIS LAI Product

We utilized the field-measured LAI values at the 500 m plot level and 30 m LAI maps to validate the proposed method. The validation process was conducted at two sites, Honghe (47°39′N, 133°31′E) and Hailun (47°24′~47°26′N, 126°47′~126°51′E), located in Heilongjiang Province, northeastern China. The crop types include rice, corn, soybean, and

sorghum. A total of 180 sets of field-measured LAI data were collected by Fang et al. in 2012, 2013, and 2016 (https://doi.pangaea.de/10.1594/PANGAEA.900090, accessed on 10 August 2021) [61,66,67]. The LAI data were collected using the LAI-2200 instrument and underwent comprehensive validation during the majority of the growing seasons. Furthermore, they were utilized to validate the precision of various crucial satellite LAI products [61]. In addition to validating the accuracy of the results in this study, the field measurements were also utilized in determining the optimal number of observation directions.

LAI maps with a spatial resolution of 30 m, obtained from a previous study [61], were used alongside the corresponding LAI measurements. These high-resolution reference LAI maps were generated using HJ-1, Landsat 7, and Sentinel-2A images and demonstrated strong agreement with the field-measured LAI values. For the purpose of cross-validation, our proposed method was valid using 284 sets of aggregated LAI values at a 1.5 km scale, derived from the 30 m LAI maps in this study. The MODIS BRDF data with spatiotemporal consistency with these measurements were collected by Zhang et al. in 2021 [30]. To ensure data integrity, high-quality BRDF data from the nearest available time were incorporated as alternative data when the BRDF had a fill value or exhibited poor quality.

The algorithm's accuracy was also validated using high-quality MODIS LAI products in two tiles. The MCD15A3H Version 6.1 MODIS Level 4 LAI product is a 4-day composite dataset with 500 m pixel size. Tile h26v04 is located in northeastern China, while tile h12v04 is situated in northeastern America. According to the statistics derived from the Annual International Geosphere-Biosphere Programme (IGBP) classification at the MODIS pixel scale, tile h26v04 is primarily characterized by grassland, comprising 53% of the tile's coverage, along with 20% cropland. On the other hand, tile h12v04 is predominantly covered by forests, with 24% Mixed Forests, 18% Deciduous Broadleaf Forests, 8% Woody Savannas, and 5% Cropland. The selected time for h26v04 is days 181–193 of the year 2020, while the selected time for h12v04 is days 245–257. The vegetation during these selected time periods corresponds to the growing season, where the vegetation exhibits thriving and robust growth. To ensure high-quality standards, it is imperative to mandate that all MODIS LAI product data within a 16-day period exhibit high quality, with the condition that the standard deviation of the four LAI values remains below 0.4. Furthermore, the reflectance values in the red and NIR bands for 397 directions were simulated using the high quality BRDF product (MCD43A1) and the kernel-driven BRDF model at the corresponding 4 days. It is required that the RMSE between the multi-angular reflectance values of different days remains below 0.01.

## 3. Results

### *3.1. Sensitivity of the PROSAIL Model to LAI and Consistency with the Kernel-Driven BRDF Model*

#### 3.1.1. Sensitive Directions to LAI Variations

The dataset of 20,000 sets of vegetation parameter combinations was expanded by setting the LAI to range from 0.5 to 10 with a step size of 0.5. Figure 2 illustrates a three-dimensional shape of simulated reflectance under different LAI parameter conditions based on the common value and PROSAIL models with a solar zenith angle of 45°. The results demonstrate significant variations in the simulated bidirectional reflectance with changing LAI values. Figure 3 presents examples of directional reflectance variations with changing LAI values, simulated using the common value and PROSAIL model in three specific directions. It shows that as LAI is low, there are significant changes in surface reflectance with the increase in LAI. However, as LAI further increases (>3.5), the reflectance tends to remain relatively constant. It also demonstrates that the intensity of LAI variations differs across different directions and spectral bands, and the σ of reflectance in a specific direction can be used to measure the magnitude of reflectance variation. In other words, directions with higher σ values indicate that the PROSAIL model is more sensitive to LAI variations.

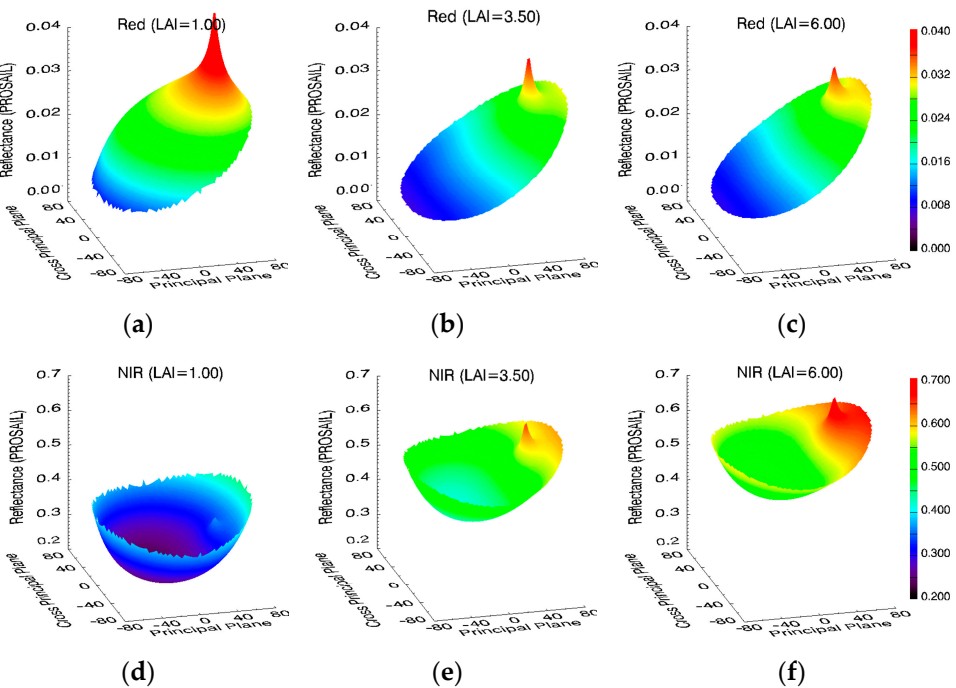

**Figure 2.** Three-dimensional BRDF shape in the red (**a**–**c**) and NIR (**d**–**f**) bands simulated based on the common value and PROSAIL model under different LAI parameter conditions. Different colors represent the magnitude of the reflectance. In the bottom coordinate plane, the radius represents the zenith angle, while the polar angle represents the azimuth angle. The vertical axis is used to plot the BRDF values.

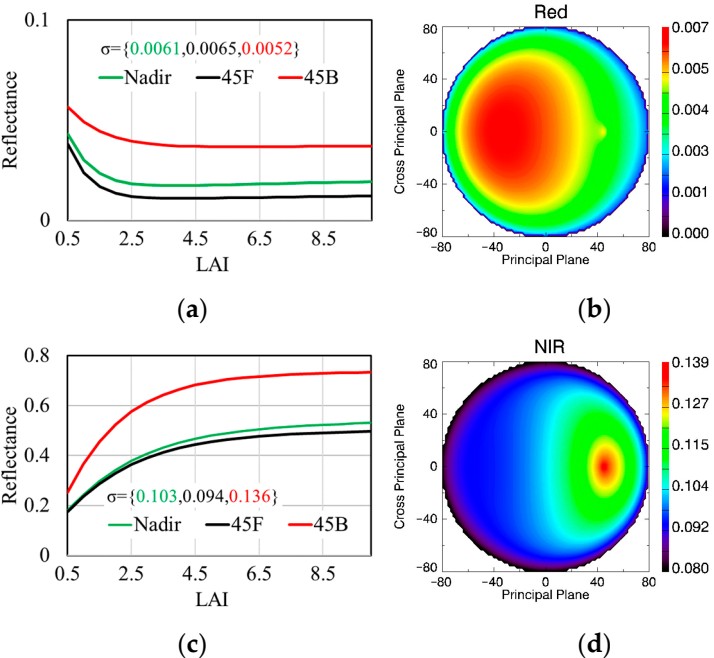

**Figure 3.** Here, (**a**,**c**) refer to the directional reflectance variations with changing LAI values, simulated using the common value and PROSAIL model in the nadir (green line), 45° forward (45F, black line) and 45° backward (45B, red line) in the red and NIR bands under a solar zenith angle of 45°. The σ of directional reflectance with changing LAI is also demonstrated. Then, (**b**,**d**) refer to the distribution of σ in the viewing hemisphere. The radius represents the zenith angle, while the polar angle represents the azimuth angle. Different colors represent the magnitude of the σ.

Figure 3 only shows an example of the distribution of σ for the common value over the viewing hemisphere. We employed a total of 20,000 sets of vegetation parameter combinations, and by calculating the average of these 20,000 sets of σ, we can determine the directions that are more sensitive to variations in LAI. The distribution of mean σ at different solar zenith angles over the viewing hemisphere is shown in Figure 4. The results indicate that in the red band, when the solar zenith angle is small, high sensitivity is observed near the small view zenith angles area. However, in the NIR band, under large solar zenith angles area, high sensitivity is observed near hotspots. The sensitivity of the PROSAIL model to changes in LAI varies across different directions in different spectral bands, highlighting the importance of considering each band individually.

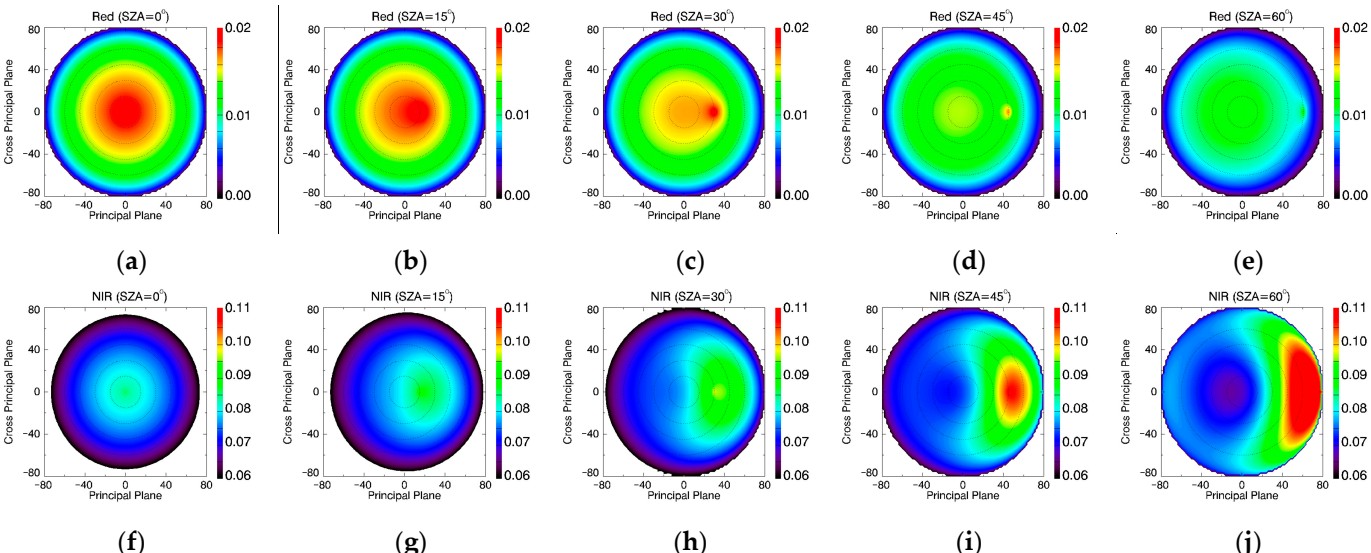

**Figure 4.** The distribution of the average σ of 20,000 sets of data over the viewing hemisphere at solar zenith angles (SZA) of 0° (**a**,**f**), 15° (**b**,**g**), 30° (**c**,**h**), 45° (**d**,**i**), and 60° (**e**,**j**) in the red (**a**–**e**) and NIR (**f**–**j**) bands.

### 3.1.2. Consistent between Two Models

The 20,000 sets of vegetation parameter combinations were sequentially inputted into the PROSAIL model to simulate the observation of hemispherical reflectance ($\rho$) at different solar zenith angles. These reflectance values were then used as inputs in the RTLSR_C BRDF model to calculate model parameters, allowing for the simulation of reflectance ($\rho_0$) in any given direction. Figure 5 shows an example of the three-dimensional BRDF shapes of $\rho_0$ based on $\rho$ simulated using the common value and PROSAIL model. Compared to Figure 2b,e, the results indicate a high level of consistency between the BRDF simulated by the RTLSR_C BRDF model and the results obtained from the PROSAIL model for the common value dataset. Comparative analysis of these two BRDF models enables a quantitative evaluation of the consistency between them.

To assess the consistency between the two models in any direction, 20,000 sets of $\rho$ and corresponding $\rho_0$ from both models are compared. Figure 6 shows the comparison between the reflectance of the two models in two directions in the NIR band when the solar zenith angle is 30°. The consistency of the two models varies significantly across the two directions, which can be quantified using RMSE. Typically, the consistency in the nadir direction, which is commonly studied, is not significant, with an RMSE of 0.042. In comparison, another direction with a view zenith angle of 40° in the principal plane exhibits a lower RMSE of 0.012, indicating a stronger consistency between the two models in that particular direction.

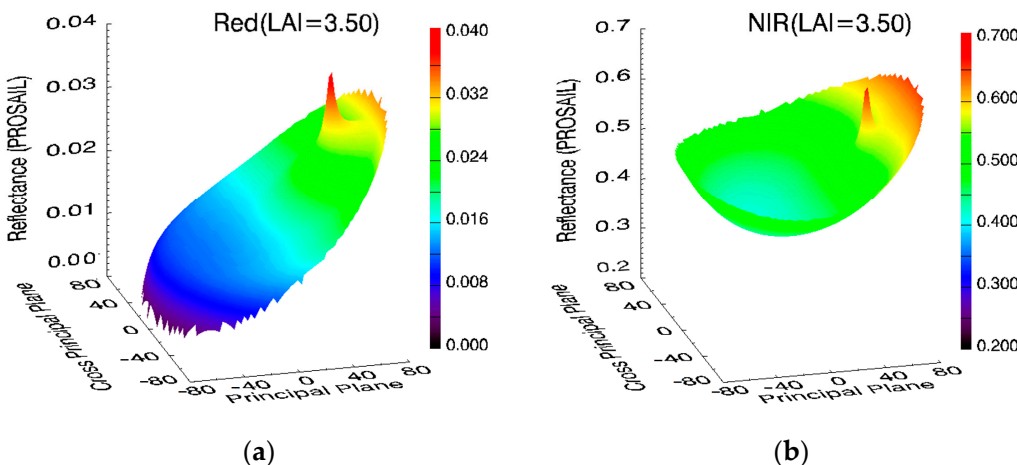

**Figure 5.** The three-dimensional BRDF shapes based on the RTLSR_C model and the multi-angular reflectance simulated using the Common Value and PROSAIL models in the red (**a**) and NIR (**b**) bands.

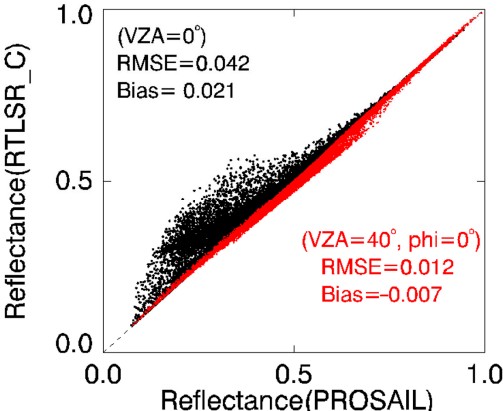

**Figure 6.** Comparison of reflectance in the NIR band between the PROSAIL and RTLSR_C models in the nadir direction (black points) and at a backscattering angle of 40° (red points) when the solar zenith angle is 30°.

By calculating the RMSE between the reflectance from the two models across various viewing directions, we can quantitatively evaluate the level of agreement between them. Figure 7 shows the distribution of the RMSE between the reflectance of the two models over the whole viewing hemisphere under different solar zenith angles. A lower RMSE indicates a higher level of consistency, suggesting that the two models yield similar results in terms of reflectance predictions. The consistent direction varies with solar zenith angle changes. Although efforts have been made to correct for the hot-spot effects, the consistency between the two models remains relatively low around the hot-spot direction. Additionally, in the red band, when the solar zenith angle is between 15° and 45°, the consistency is poor in some directions within the forward hemisphere. On the other hand, in the NIR band, when the solar zenith angle is greater than 30°, there is a weaker correlation near the nadir direction.

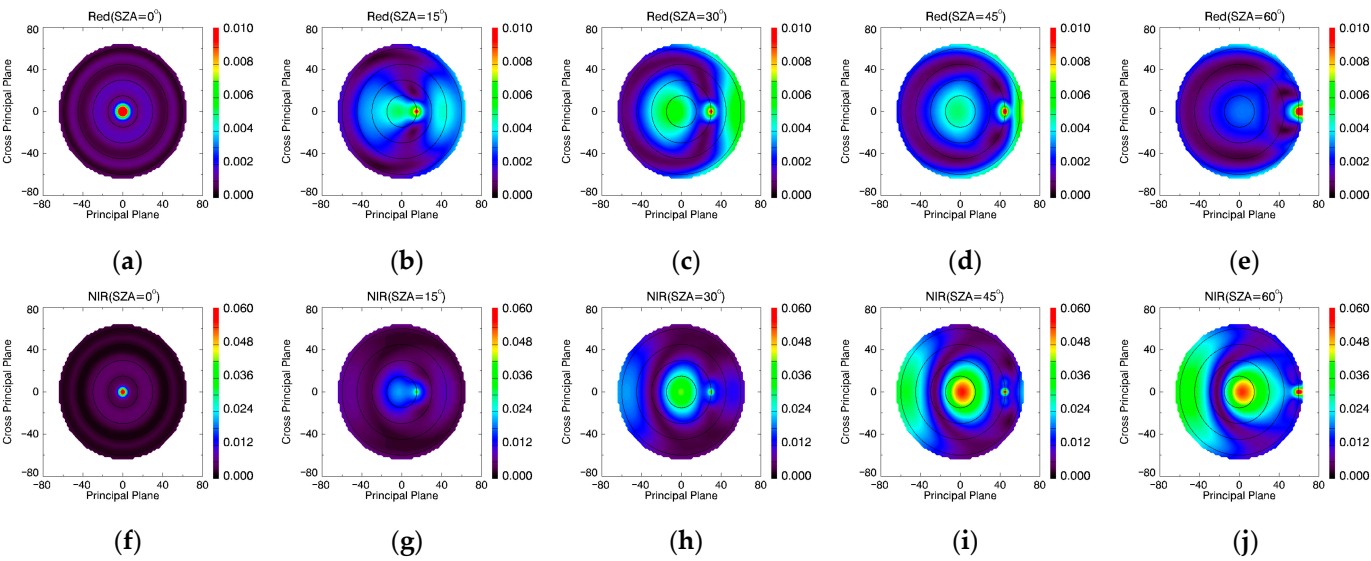

**Figure 7.** The distribution of the RMSE between the reflectance from the PROSAIL and RTLSR_C models over the whole viewing hemisphere at solar zenith angles of 0° (**a**,**f**), 15° (**b**,**g**), 30° (**c**,**h**), 15° (**d**,**i**), and 60° (**e**,**j**) in the red (**a**–**e**) and NIR (**f**–**j**) bands.

### 3.2. Optimal Observation Geometry for LAI Retrieval

To jointly utilize these two models for LAI inversion, it is necessary to select directions that are sensitive to LAI variations and exhibit high consistency between the two models. Based on the previous analysis using 397 directions, and considering the sensitivity of the PROSAIL model to LAI changes and model consistency, the optimized observation directions can be selected by adjusting the threshold. To determine the optimal number of observations for LAI retrieval, different numbers of observations were selected and studied, and ultimately the optimal number of observations was determined. The σ and RMSE thresholds set for different bands and observation numbers are shown in Table 1.

**Table 1.** The σ and RMSE thresholds set for the red and NIR bands with respect to different observation numbers.

| No. | Red | | NIR | |
|---|---|---|---|---|
| | σ | RMSE | σ | RMSE |
| 15 | 0.0235 | 0.0038 | 0.0860 | 0.0148 |
| 30 | 0.0220 | 0.0040 | 0.0800 | 0.0150 |
| 60 | 0.0201 | 0.0040 | 0.0750 | 0.0150 |
| 90 | 0.0181 | 0.0041 | 0.0676 | 0.0152 |
| 150 | 0.0145 | 0.0045 | 0.0500 | 0.0325 |

Figure 8 illustrates the distribution of observations in the red and NIR bands for 30, 60, and 90 selected observations. Considering both models have a symmetrical distribution about the principal plane, only half of the observation hemisphere's viewing directions were taken into account. Similar to the previous findings, the data in the red band are concentrated in observation directions with smaller solar zenith angles. However, in the NIR band, there is a larger proportion of observations with larger solar zenith angles.

Once the observation directions were determined, the reflectance for various bands in those directions was simulated using the PROSAIL model. By incorporating the input vegetation parameters, a new lookup table was generated to establish the connection between the reflectance at different observation directions and the vegetation parameters. This lookup table will be utilized in the subsequent LAI inversion, which is based on a MODIS BRDF product.

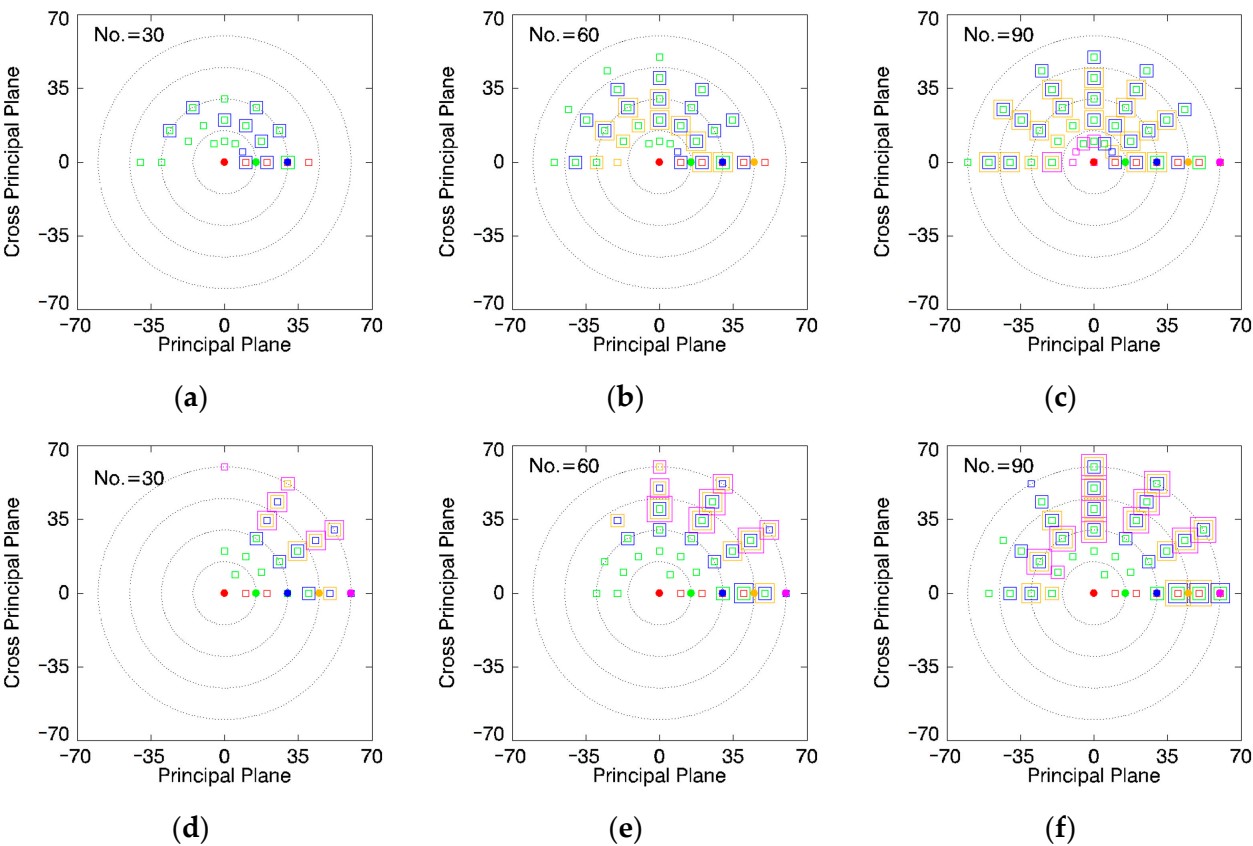

**Figure 8.** The distribution of observations in the red (**a**–**c**) and NIR (**d**–**f**) bands for 30 (**a**,**d**), 60 (**b**,**e**), and 90 (**c**,**f**) selected directions. Solid circles represent the position of the sun, while hollow rectangles represent the observation locations. The colors red, green, blue, gold, and purple correspond to observations taken at solar zenith angles of 0°, 15°, 30°, 45°, and 60°, respectively. Different-sized rectangles are used to indicate the positions of the observation data in order to avoid overlap.

*3.3. Validation of the LAI Estimations Using Field Measurements and LAI Maps*

A total of 180 field LAI measurements at the 500 m plot and the high-resolution (30 m) LAI maps were utilized for algorithm validation. High-quality MODIS BRDF products corresponding to the spatiotemporal coverage of these surface observations were employed for LAI inversion. In cases where the MODIS BRDF product quality was low, it was replaced with the nearest neighboring high-quality BRDF product to ensure data integrity.

During the LAI inversion process, the reflectance for selected observation directions was first simulated based on the MODIS BRDF product. Then, the simulated MODIS reflectance was compared with the reflectance in the new lookup table. To limit the size of the lookup table and improve the efficiency of the inversion process, the empirical ALA was calculated using the $f_{vol}$-ALA relationship when the $f_{vol}$ in the NIR band was less than 0.3813 [30]. Finally, the RMSE$_c$ was calculated between the simulated MODIS reflectance and the reflectance values in the lookup table. The average LAI value corresponding to the 50 sets of data with the lowest RMSE$_c$ was considered to be the inversion LAI.

Figure 9 shows the comparison of 180 LAI measurements in the 500 m plot with LAI estimated from MODIS BRDF. Among them, Figure 9a is the result of previous research provided by Zhang et al., 2021 [30], using 397 observations. After the sensitivity analysis and consistency analysis, the inversion results based on 30 directions had a similar accuracy (RMSE = 1.37, Bias = −0.16, R$^2$ = 0.43) to the result of Figure 9a. In comparison, the coefficient of determination (R$^2$) slightly decreased, the RMSE slightly increased, but the absolute magnitude of bias decreased. Despite the significant reduction in the number of observations, the inverted LAI still exhibited a high level of consistency with the surface

observation data, and also exhibited higher accuracy compared to the MODIS LAI product generated with the main algorithm, as evidenced by the evaluation metrics: RMSE of 1.50, bias of −0.23, and an $R^2$ value of 0.24 [68]. Figure 9c shows the $R^2$ between the results obtained based on different numbers of observations and surface observation data. The results indicate that when reducing the number of observations, the highest consistency with ground observations is achieved through inversion using 30 directions. In the following research, the retrieval of LAI will focus on utilizing the new lookup table specifically designed for the case with 30 observations.

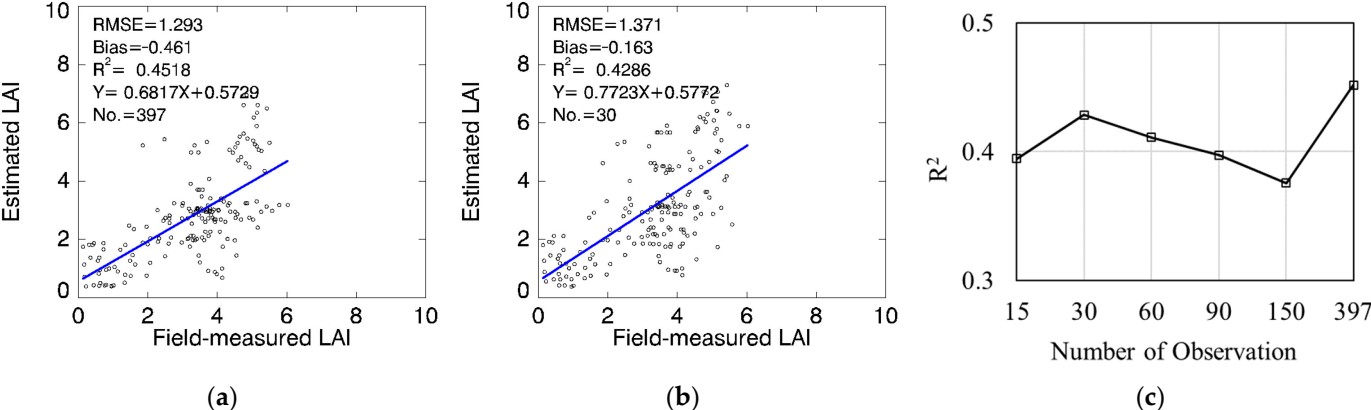

**Figure 9.** The comparison of 180 LAI measurements at the 500 m plot with LAI estimated from MODIS BRDF based on different numbers of observations. Among them, (**a**) is the result based on 397 observations; (**b**) is the result based on 30 directions; and (**c**) is the $R^2$ between the results obtained based on different numbers of observations and surface observation data. The blue line in (**a**,**b**) represents the fitted relationship between these two sets of data.

The 30 m high-quality LAI maps were initially aggregated to a 500 m scale, and the inverted LAI values were also obtained at the 500 m resolution. The comparison of 30 m high-quality LAI maps was conducted at a scale of 1.5 km (3 × 3 MODIS pixels), resulting in a total of 284 data sets. Figure 10a is the result of previous research provided by Zhang et al., 2021 [30], using 397 observations per band; Figure 10b is the result based on 30 observations. Similar to the previous findings, despite the reduction in the number of observations used for inversion, the accuracy of the inversion results was not affected and there was a better agreement between the retrieved LAIs and the LAI maps.

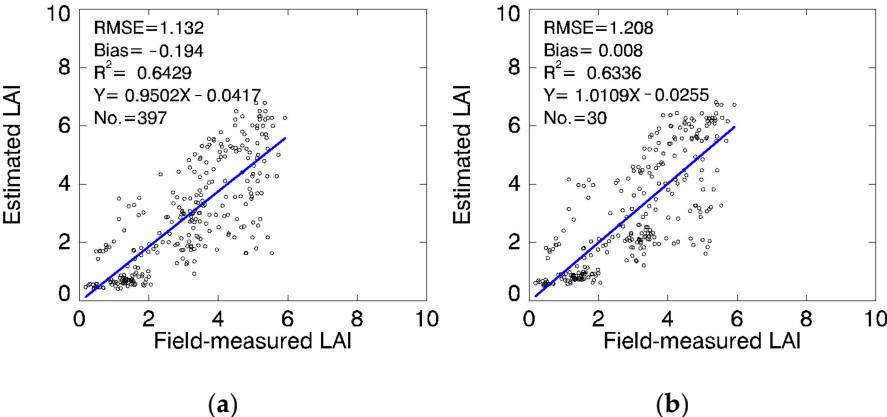

**Figure 10.** The comparison of 30 m high-quality LAI maps with LAIs retrieval from MODIS BRDF at a scale of 1.5 km: (**a**) is the result based on 397 observations provided by Zhang et al., 2021, [10] and (**b**) is the result based on 30 selected observations. The blue line represents the fitted relationship between these two sets of data.

### 3.4. Validation of LAI Estimation Based on MODIS LAI Product

To filter the selected MODIS BRDF data, it is necessary to ensure that both datasets are of high quality. Additionally, the LAI product within the 16-day period should exhibit stability, and the variations in the 397 directional reflectance data simulated by the MODIS BRDF model should be minimal. The choice of 397 directions instead of 30 directions is primarily aimed at facilitating a comparison between the proposed method and the previous approach.

The comparison between LAIs retrieval using MODIS BRDF and the MODIS LAI product is shown in Figure 11. For tile h26v04, the LAI values are relatively small, with the majority of pixels having an LAI less than 3.5. In the case of tile h12v04, the MODIS LAI values are divided into two parts: one part is below 3.5, while the other part is concentrated around 5. Compared to the inversion results based on 397 angles, the results based on 30 observation angles exhibit higher consistency. The RMSE of the two scenes decreases from 0.354 to 0.340 and from 2.424 to 2.038, respectively. The results also demonstrate that the LAI accuracy obtained from simulated multi-angle data inversion is relatively high when the LAI is less than 3.5. However, when the LAI exceeds 4.5, the LAI estimated from multi-angle data inversion tends to be significantly underestimated. This can be attributed to the use of the look-up table method, the uniform distribution of vegetation parameters employed in the PROSAIL model, and the reflectance simulated by the PROSAIL model, which shows relatively small variations when LAI is greater than 3.5. The mean value of LAI corresponding to the 50 sets of minimum $RMSE_c$ data can cause the inversion results to underestimate LAI in higher LAI areas.

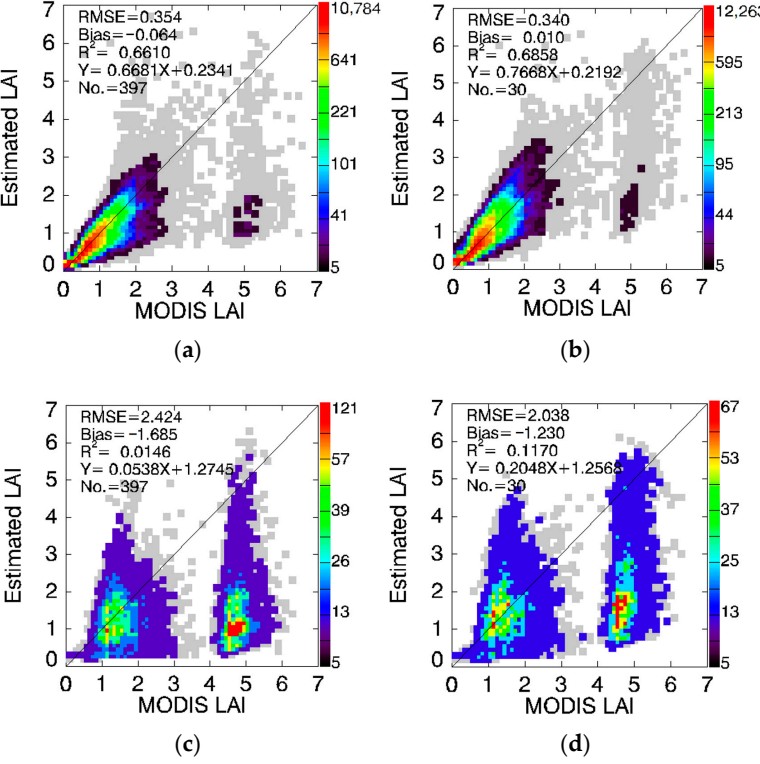

**Figure 11.** The comparison between LAI retrieval using MODIS BRDF and the MODIS LAI product. Specifically, (**a**,**b**) show the results for tile h26v04 during days 181–193 of the year 2020, while (**c**,**d**) display the results for tile h12v04 during days 245–257 of the year 2020. Here, (**a**,**c**) refer to LAI retrieval from 397 directional reflectance per band, while (**b**,**d**) refer to LAI retrieval from 30 directional reflectance per band. Different colors represent the number of pixels in each category, with gray indicating a pixel number of less than 5.

## 4. Discussion

The accurate inversion of LAI not only plays a crucial role in advancing ecological research but also holds practical applications in improving agricultural productivity and maintaining the health of vegetation. In this paper, we examined the improvement in LAI estimation by utilizing multi-angle reflectance data and the PROSAIL model, with the application of MODIS BRDF products and an enhanced lookup table method. The results indicate that by selecting highly sensitive and consistent directions, even though the number of observation directions is reduced, the improved lookup table based on surface observations can still achieve high accuracy in LAI retrieval. Additionally, reducing the number of observation directions significantly improves the efficiency of inversion. This study also provides new insights for the coupled inversion of surface parameters using both physical and semi-empirical kernel-driven models.

The anisotropic reflectance characteristics of the Earth's surface play a crucial role in the inversion of surface parameters, as they provide valuable information about its structural features [46]. The physical BRDF model accurately describes the surface reflection behavior. By incorporating detailed vegetation structure descriptions, this model enables the simulation of directional reflectance for various parameter combinations [60]. Through an inversion strategy, the vegetation structure can be estimated using reflectance data. Numerous studies have successfully inverted vegetation with high accuracy by leveraging observed multi-angle reflectance [31,53]. Coupling the kernel-driven and PROSAIL models by analyzing the consistency between the models as well as their sensitivity to land surface structural parameters enables the creation of a lookup table that establishes the correlation between specific direction reflectance and land surface parameters. In order to fully utilize the surface's anisotropic reflection characteristics and provide a simplified LAI retrieval method, this study couples the PROSAIL BRDF model with the RTLSR_C model to invert LAI. Although there are discrepancies between the models, a previous study demonstrated a strong agreement in BRDF when integrating the two models, even when the PROSAIL input parameters underwent substantial variations [57]. This serves as a fundamental basis for the current study.

The most significant improvement compared to previous studies is that this study searched for the optimal number and positions of viewing angles that are needed to obtain the best inversion results. By examining the changes in directional reflectance simulated by the PROSAIL model as a function of LAI, the directions that are sensitive to changes were identified. By inputting the output of the PROSAIL model into the kernel-driven model, the consistency between the two models was explored. This process helped to determine the optimal directions and provided a basis for selecting the observation directions. In general, the optimal directions tend to exhibit a clustered distribution, meaning that under different solar zenith angles, the more sensitive and highly correlated directions are often concentrated within a smaller range. Considering the strong correlation of the model-simulated reflectance in adjacent directions, the study evaluated the existing discrete distribution of 397 observation directions [30], fully considering the surface's anisotropic information. It is worth noting that although the improved RTLSR_C model captured the bidirectional reflectance characteristics of the hotspot, the 30 selected directions did not include the reflectance near the hotspot. This is because there were still significant differences between the two models in the surroundings of the hotspot area [54], and further research is needed to improve the characterization of the hotspot.

This study is based on high-quality MODIS 500 m BRDF products for LAI inversion. The improved algorithm significantly enhances computational efficiency. In comparison to LAI derived from surface measurements, it achieves similar levels of accuracy with an algorithm based on 397 observations [30]. Compared to the MODIS LAI product, the retrieved LAI results demonstrate better consistency with both the LAI measurement and the LAI maps. This also indicates that multi-angle observation data can provide advantageous assistance in improving the accuracy of LAI retrieval [40,69]. For high-resolution remote sensing data that have multi-angle observations, these simulated BRDF

data from the RTLSR_C model hold promise as compensatory means. Certainly, further validation of this method is needed with additional surface measurements.

Based on the validation of MODIS LAI products, the improved lookup table method shows relatively better performance when LAI is less than 3.5. There are certain improvements in terms of correlation coefficient and RMSE. However, for larger LAI values, although there is some improvement, both methods yield relatively poor results. There are several reasons that contribute to this phenomenon. Firstly, the radiative transfer models like PROSAIL consist of a set of input variables that always appear in combination, making the inversion of PROSAIL an ill-posed problem [70]. This could be a common issue with lookup table methods. The results are based on the average of the 50 data sets with the lowest RMSE, which can lead to a significant underestimation of LAI values, particularly when the LAI is large. Another possible reason is that this issue is also influenced by the configuration of model parameters. In the study, the parameter combinations used were uniformly distributed [30], which did not accurately represent the actual surface characteristics. Since each parameter has an impact on the final BRDF and can introduce deviations in the results, it is crucial to optimize the parameter combinations based on the specific vegetation features of the study area [31]. This optimization process would further enhance the accuracy of the inversion results. In addition, it is important to consider that the coarse resolution of MODIS limits its ability to fully capture the detailed surface parameter characteristics of a pixel with an average parameter combination [71]. The simulated dataset based on MODIS BRDF does not fully capture the complexity and diversity of the actual land surface. Further research can explore higher-resolution BRDF data and more complex canopy structures to better understand the potential advantages of multi-angle reflectance data in LAI estimation.

In summary, the use of the improved lookup table approach, compared to the original algorithm, enables the maintenance of high LAI accuracy while improving the efficiency of the inversion process. The estimation process described in this study can be widely expanded to incorporate a range of semi-empirical kernel driven and physical BRDF models for extracting various surface parameters in large, homogeneous regions. The next step will involve optimizing the distribution of vegetation parameters based on surface characteristics and further validating the method by incorporating high-resolution BRDF data obtained from surface measurements. This will provide a more comprehensive validation of the approach and enhance the accuracy of LAI estimation.

## 5. Conclusions

Coupling the kernel-driven BRDF model with the PROSAIL model, and utilizing the reflectance in directions that are sensitive to LAI changes, while maintaining high consistency between two models, can improve the accuracy of LAI retrieval. In this study, we focused on determining the optimal number and positions of viewing angles required to enhance the inversion results for LAI extraction using the high-quality MODIS BRDF product and PROSAIL model. By utilizing 20,000 sets of surface vegetation parameter combinations, we explored the sensitivity of the PROSAIL model to variations in LAI and its consistency with an RTLSR_C model. Through the application of a threshold, we identified the optimal viewing directions for LAI retrieval. According to the validation using field-measured LAI data, it was found that the best results for LAI retrieval were obtained when 30 viewing angles were selected in the inversion using the MODIS BRDF model. The proposed method was further validated using field measurements and high-quality, high-resolution LAI maps, which indicated that the inversion results were comparable in accuracy to previous algorithms based on 397 viewing angles. Validation using two tiles of high-quality MODIS LAI products revealed that the proposed method exhibited a high level of consistency with the MODIS LAI product when the LAI values were less than 3.5. However, in forest-dominated regions where LAI values were higher, although there was some improvement compared to previous methods, the extent of improvement was relatively weak. This may be attributed to the PROSAIL model's ability to capture the

anisotropic characteristics of the Earth's surface, the algorithm's utilization of uniformly distributed input parameters, and the minimal variations in PROSAIL-simulated reflectance at high LAI.

In the future, unmanned aerial vehicles will be employed to acquire high-resolution surface BRDF datasets. The focus will be on redesigning the distribution of vegetation parameter combinations based on the actual characteristics of surface vegetation. Furthermore, the algorithm will be enhanced by integrating the kernel-driven BRDF model with additional physical models, aiming to improve the inversion accuracy in regions with high LAI values.

**Author Contributions:** Conceptualization, H.Z. and Y.L. (Yan Liu); methodology, H.Z. and X.Z.; validation, Y.L. (Yi Lian) and H.C.; formal analysis, L.C. (Lei Cui); investigation, Y.D.; data curation, L.C. (Lei Chen), X.Z. and Q.X.; writing—original draft preparation, H.Z.; writing—review and editing, Y.L. (Yan Liu), X.Z. and Y.D.; funding acquisition, H.Z. and Y.L. (Yan Liu). All authors have read and agreed to the published version of the manuscript.

**Funding:** This work was supported by the National Natural Science Foundation of China (41971306).

**Data Availability Statement:** All satellite remote sensing and field measured data used in this study are openly and freely available. LAI measurements at the 500 m plot level and 30 m LAI maps can be accessed via https://doi.pangaea.de/10.1594/PANGAEA.900090 (accessed on 10 August 2021). The Collection 6 MODIS BRDF parameter product (MCD43A1 and MCD43A2) is available at https://search.earthdata.nasa.gov/search (accessed on 10 March 2023).

**Acknowledgments:** We express our sincere gratitude to the research team of Hongliang Fang at the Chinese Academy of Sciences for generously sharing their LAI measurements as public data. Our gratitude also goes to the PROSAIL team for making their model code freely accessible online. Additionally, we appreciate the research team of the MODIS BRDF products for freely providing global data. Lastly, we are thankful for the thorough review and valuable feedback provided by the anonymous reviewers.

**Conflicts of Interest:** The authors declare no conflict of interest.

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
