# Peer review of "Enhancing Leaf Area Index Estimation with MODIS BRDF Data by Optimizing Directional Observations and Integrating PROSAIL and Ross–Li Models"

_remotesensing, doi:10.3390/rs15235609_

Round 1
Reviewer 1 Report
Comments and Suggestions for Authors
I think this is a decent attempt to introduce a strategy to obtain the optimal viewing angles of MODIS BRDF data to estimate LAI. The new method will be helpful to enhance LAI estimation from MODIS BRDF data. Furthermore, the overall logic of the manuscript is clear. Therefore, I would recommend a minor revision.
Detailed comments:
L134-138: Please rephrase this sentience, too long.
L146: The frame does not need to be expressed in different colors, please unify it into black.
L211-214: Here are some rules to divide the range of SZA and VZA, I just wonder how to choose the spacing, and if these spacings are best for your method.
L217: The reason why using the average value of σ can ensure a more representative outcome is not clear, please explain it.
234-235: Please rephrase this sentience.
L247 and 249: Please do double-check for these two sentences, just make sure the expression of р is correct.
L258: What is “the previously adopted strategy”, it is not clear. Please rephrase this sentence.
L261: In this part, I suggest you use a table to introduce your used datasets, simplify the description, and retain important information about the data used, such as time, accuracy, etc.
L316: Enhance the quality of Fig.2 such as the axes name.
L340: In your method, the lookup table is constructed using simulated data from PROSAIL, but the actual inversion uses MODIS data. You have conducted the consistency evaluation of these two models to make sure your look-up-table is reliable. However, you found a poor consistency in the hotspot direction, will it affect the inversion accuracy of your method?
L442:Fig.9 and Fig. 10, please adjust the range of the coordinate axis.
L456: In comparison with MODIS LAI products, it can be found that there is still a big difference in accuracy overall. Although the inversion of LAI products is carried out based on the same observation data source, where is the problem with such differences? Please further analyze or explain. I think this might help readers better.
Author Response
Point-to-point responses to the reviewer's comments #1.
Comments and Suggestions for Authors
I think this is a decent attempt to introduce a strategy to obtain the optimal viewing angles of MODIS BRDF data to estimate LAI. The new method will be helpful to enhance LAI estimation from MODIS BRDF data. Furthermore, the overall logic of the manuscript is clear. Therefore, I would recommend a minor revision.
Reply: Thank you very much for your positive comments and precious suggestions that helped to improve the manuscript. We revised our manuscript and responded to the comments and suggestions as follows.
Detailed comments:
L134-138: Please rephrase this sentience, too long.
Reply: Thanks for your comment.
We rephrase this sentence as: “The first component focuses on analyzing the PROSAIL model's sensitivity to variations. The second component investigates the coherence between PROSAIL and RTLSR_C BRDF models under different observation geometries. The third component involves using the results obtained from the previous two steps to determine the optimal direction based on the analysis conducted with the 397 viewing directions.”
L146: The frame does not need to be expressed in different colors, please unify it into black.
Reply: Thanks for your comment. We have made revisions as suggested.
L211-214: Here are some rules to divide the range of SZA and VZA, I just wonder how to choose the spacing, and if these spacings are best for your method.
Reply: Thanks for your comment.
We referenced previous studies to divide the observation geometry. Reflectance with similar observation geometries exhibits a higher correlation. By dividing the entire space with regular intervals, we aim to minimize reflectance similarities while adequately accounting for the anisotropic reflectance characteristics of the Earth's surface.
L217: The reason why using the average value of σ can ensure a more representative outcome is not clear, please explain it.
Reply: Thanks for your comment.
σ refers to the standard deviation of reflectance derived from PROSAIL model and different parameter combinations in a specific direction. A larger value of σ indicates a higher sensitivity. This study uses 20000 sets of combinations, therefore, with 20,000 values for σ in each direction, averaging is a good choice to assess the sensitivity of reflectance to changes in LAI for a specific direction.
234-235: Please rephrase this sentience.
Reply: Thanks for your comment.
We rephrase this sentence as “The RMSE is computed for each direction using ρ and ρâ‚€, with a value of k set at 20,000. A lower RMSE value signifies greater consistency between the two models.”
L247 and 249: Please do double-check for these two sentences, just make sure the expression of р is correct.
Reply: Thanks for your comment.
We confirm that the expressions in these two sentences are correct. ρ represents the simulated directional reflectance based on PROSAIL model while ρ0 represents the corresponding simulated data from MODIS BRDF.
L258: What is “the previously adopted strategy”, it is not clear. Please rephrase this sentence.
Reply: Thanks for your comment.
This sentence intends to convey that, by adopting the strategy used in previous research, BRDF data that have fvol outside the range can also be used to retrieval LAI. We modify this sentence to “To ensure that all data can participate in the calculation,”
L261: In this part, I suggest you use a table to introduce your used datasets, simplify the description, and retain important information about the data used, such as time, accuracy, etc.
Reply: Thanks for your comment.
Here, three types of LAI data—field-measured LAI, LAI maps, and high-quality MODIS LAI—are mentioned for validating the LAI accuracy extracted by the algorithm in this study. The article is structured into three paragraphs to discuss the time of data acquisition, locations, resolution, and the usage context of the data. There are substantial differences among the three types of data, and a simple table cannot provide a clear description of these distinctions.
L316: Enhance the quality of Fig.2 such as the axes name.
Reply: Thanks for your comment.
We have modified the Fig.2 to the following style:
(a) |
(b) |
(c) |
(d) |
(e) |
(f) |
L340: In your method, the lookup table is constructed using simulated data from PROSAIL, but the actual inversion uses MODIS data. You have conducted the consistency evaluation of these two models to make sure your look-up-table is reliable. However, you found a poor consistency in the hotspot direction, will it affect the inversion accuracy of your method?
Reply: Thanks for your comment.
The consistency of the kernel-driven BRDF model in the hotspot direction with the PROSAIL model remains relatively poor after adjusting the hotspot. Although the current approach has some improvements for hotspot, further research is still needed. The hotspot correction direction used in this study only alters the reflectance near the hotspot, with little impact on areas far from the hotspot area. In addition to considering the consistency between the two models, the method proposed in this study also needs to consider the sensitivity of the models to LAI changes. Given the current research foundation, the hotspot direction may not meet the conditions. Reflectance in multiple directions can capture the anisotropic reflectance characteristics of the Earth's surface, providing a more accurate estimation of LAI.
L442:Fig.9 and Fig. 10, please adjust the range of the coordinate axis.
Reply: Thanks for your comment.
We consider setting the max value of LAI to 10 for the two figures is reasonable. Firstly, when designing the lookup table, we defined the LAI range as 0-10. Additionally, this range has been used in previous study by Zhang et al., 2021. When setting a smaller range, some points would be obscured by text. Therefore, we did not make modifications to these two figures.
L456: In comparison with MODIS LAI products, it can be found that there is still a big difference in accuracy overall. Although the inversion of LAI products is carried out based on the same observation data source, where is the problem with such differences? Please further analyze or explain. I think this might help readers better.
Reply: Thanks for your comment.
In section 3.4 we identified the potential causes of such differences. In the revised version, we added one more situation that may lead to this issue. Figure 2 shows that when LAI is low, with the increase of LAI, the surface reflectance undergoes significant changes. However, with a further increase in LAI (>3.5), the reflectance remains relatively constant. Therefore, when the LAI is high, the inversion results are comparatively poorer. We add this possible reason into the revised manuscript. the content is as follows: “This can be attributed to the use of the look-up table method, the uniform distribution of vegetation parameters employed in the PROSAIL model and the reflectance simulated by the PROSAIL model shows relatively small variations when LAI is greater than 3.5.”
Once again, we appreciate your insightful suggestions.

Reviewer 2 Report
Comments and Suggestions for Authors
This manuscript presents an interesting approach for improving LAI estimation. The utilization of specific directions and the coupling of the RTLSR_C and PROSAIL models offer a novel perspective on this topic. The methodology employed in the study, particularly the analysis of a dataset comprising 20,000 vegetation parameter combinations, adds robustness to the research. The identification of directions with higher sensitivity to LAI changes and better consistency with the Ross-Li BRDF models provides valuable insights. The manuscript contributes to the advancement of LAI estimation techniques and offers practical implications for large-scale applications. Before the article can be published, there are still the following issues that need to be addressed.
1. The manuscript would benefit from a more comprehensive discussion of the implications and significance of the proposed method. While the results and validation are presented, it would be valuable to further elaborate on the potential applications and benefits of accurate LAI estimation, such as improved understanding of ecosystem dynamics, better crop yield predictions, or enhanced monitoring of vegetation health. Providing a clear and compelling argument for the practical importance of the research would strengthen the impact of the manuscript.
2. The manuscript could benefit from a clearer presentation of the methodology used in the analysis of the dataset. While it is mentioned that the dataset comprises 20,000 vegetation parameter combinations, the specific details regarding the selection criteria, data preprocessing steps, and any assumptions made should be provided. This would allow readers to better understand the robustness and generalizability of the findings and enable other researchers to replicate or build upon the study.
3. Insufficient discussion on the experimental design. The article lacks a thorough discussion on the rationale and effectiveness of the experimental design. Provide detailed information on the experimental design for readers to assess the reliability and applicability of the experiment.
4.The discussion of the limitations and potential challenges of the proposed method in the manuscript is somewhat limited. To enhance the reliability and practical applicability of the research, it is recommended to provide a more detailed discussion on potential sources of error, uncertainties, and other factors that may influence the results.
Author Response
Point-to-point responses to the reviewer's comments #2.
The reviewer's comments and suggestions has been underlined.
This manuscript presents an interesting approach for improving LAI estimation. The utilization of specific directions and the coupling of the RTLSR_C and PROSAIL models offer a novel perspective on this topic. The methodology employed in the study, particularly the analysis of a dataset comprising 20,000 vegetation parameter combinations, adds robustness to the research. The identification of directions with higher sensitivity to LAI changes and better consistency with the Ross-Li BRDF models provides valuable insights. The manuscript contributes to the advancement of LAI estimation techniques and offers practical implications for large-scale applications.
Reply: Thank you very much for your positive comments and precious suggestions that helped to improve the manuscript. We revised our manuscript and responded to the comments and suggestions as follows.
Before the article can be published, there are still the following issues that need to be addressed.
1. The manuscript would benefit from a more comprehensive discussion of the implications and significance of the proposed method. While the results and validation are presented, it would be valuable to further elaborate on the potential applications and benefits of accurate LAI estimation, such as improved understanding of ecosystem dynamics, better crop yield predictions, or enhanced monitoring of vegetation health. Providing a clear and compelling argument for the practical importance of the research would strengthen the impact of the manuscript.
Reply: Thank you for your valuable feedback on our manuscript.
We indeed recognize the importance of providing a more comprehensive discussion on the implications and significance of the proposed method. In the first paragraph of the introduction, we have highlighted the significance of accurately retrieving LAI. In the revision, at the beginning of the discussion section, we further elaborate in the discussion section on the potential applications and benefits of accurate LAI estimation.
In the revised version, the content is as follows: “The accurate inversion of LAI not only plays a crucial role in advancing ecological research but also holds practical applications in understanding of ecosystem dynamics, improving agricultural productivity, and maintaining the health of vegetation.”
Once again, we appreciate your insightful suggestions.
2. The manuscript could benefit from a clearer presentation of the methodology used in the analysis of the dataset. While it is mentioned that the dataset comprises 20,000 vegetation parameter combinations, the specific details regarding the selection criteria, data preprocessing steps, and any assumptions made should be provided. This would allow readers to better understand the robustness and generalizability of the findings and enable other researchers to replicate or build upon the study.
Reply: Thank you for your valuable feedback on our manuscript.
The 20,000 vegetation parameter combinations were used by Zhang et al., 2021. These combinations were generated using the Satellite periodic function through uniform sampling of seven leaf and canopy parameters.
In Section 2.1, there is an introduction to the data.“This study utilizes a comprehensive simulation dataset of 20,000 vegetation parameter combinations, which was used in a previous study [30]. These combinations were generated using the Satellite periodic function through uniform sampling of seven leaf and canopy parameters, i.e., leaf structure parameter (Ns), chlorophyll a and b content (Cab), equivalent water thickness (Cw), leaf mass per unit leaf area (Cm), leaf area index (LAI), average leaf angle (ALA), and soil coefficient (Psoil). Reasonable ranges have been set for all seven parameters, and the rest of the parameters have been set to the constant values. The specific information about these leaf and canopy parameters can refer to Zhang et al., 2021 [30].”
Once again, we appreciate your insightful suggestions.
3. Insufficient discussion on the experimental design. The article lacks a thorough discussion on the rationale and effectiveness of the experimental design. Provide detailed information on the experimental design for readers to assess the reliability and applicability of the experiment.
Reply: Thank you for your valuable feedback on our manuscript.
Figure 1. Provides a detailed introduction to the experimental design of this study. In the revised version, the content is as follows: “The first component focuses on analyzing the PROSAIL model's sensitivity to variations. The second component investigates the coherence between PROSAIL and RTLSR_C BRDF models under different observation geometries. The third component involves using the results obtained from the previous two steps to determine the optimal direction based on the analysis conducted with the 397 viewing directions. A new lookup table that captures the relationship between vegetation parameters and reflectance in the optimal observation direction using the PROSAIL model is created. The fourth component involves simulating the reflectance in the optimal observation direction based on MODIS BRDF products. By comparing the simulated reflectance and the corresponding reflectance in the new lookup table, the minimum cost function is calculated to determine the LAI value. The results will be validated using ground-based LAI measurements and MODIS LAI products.”
Once again, we appreciate your insightful suggestions.
4.The discussion of the limitations and potential challenges of the proposed method in the manuscript is somewhat limited. To enhance the reliability and practical applicability of the research, it is recommended to provide a more detailed discussion on potential sources of error, uncertainties, and other factors that may influence the results.
Reply: Thanks for your comment.
We add one situation that may lead to this issue is evident from the simulation results based on the PROSAIL model.
In section 3.4 of the revised version, the revised content is as follows: “This can be attributed to the use of the look-up table method, the uniform distribution of vegetation parameters employed in the PROSAIL model and the reflectance simulated by the PROSAIL model shows relatively small variations when LAI is greater than 3.5.”
In the conclusions section, the latter part of the first paragraph has been modified as follows: “This may be attributed to the PROSAIL model's ability to capture the anisotropic characteristics of the Earth's surface, the algorithm's utilization of uniformly distributed input parameters, and the minimal variations in PROSAIL-simulated reflectance at high LAI.”
Once again, we appreciate your insightful suggestions.

Reviewer 3 Report
Comments and Suggestions for Authors
Dear Authors,
The manuscript is interesting. I highly appreciate the results obtained. I definitely recommend it for publication.

Author Response
Point-to-point responses to the reviewer's comments #3.
Comments and Suggestions for Authors
Dear Authors,
The manuscript is interesting. I highly appreciate the results obtained. I definitely recommend it for publication.
Reply: Thank you for the reviewer's affirmation and recommendation.
We have revised the conclusion section to emphasize our findings and highlight potential sources of error. Additionally, we have standardized the format of DOIs for the corresponding references.
Once again, we appreciate your insightful suggestions.

Reviewer 4 Report
Comments and Suggestions for Authors
The topic is interesting and challenging at the same time. I understand this research has approached a very hard task and seems to successfully deliver the aim. The paper is well-written, and the results are interesting. One of the notable strengths of the paper lies in its thorough literature review, which provides a solid foundation for understanding the LAI/models.
The paper starts well and in the second paragraph, the storyline is well followed. Nevertheless, at the beginning of the third paragraph, I felt almost lost and while narrowing down the topic, the paper did not consider different aspects of the topic well. I believe at the beginning of the literature a bigger picture of measurement of leaf area could be provided before abruptly moving to the models. While looking at the literature I found the following papers might be useful for a wider picture of the topic at the beginning. I hope they help.
Rahimikhoob, H., Delshad, M., & Habibi, R. (2023). Leaf area estimation in lettuce: Comparison of artificial intelligence-based methods with image analysis technique. Measurement, 222, 113636.
Mora, M., Avila, F., Carrasco-Benavides, M., Maldonado, G., Olguín-Cáceres, J., & Fuentes, S. (2016). Automated computation of leaf area index from fruit trees using improved image processing algorithms applied to canopy cover digital photograpies. Computers and Electronics in Agriculture, 123, 195-202.
Zhang, X., Jiao, Z., Zhao, C., Yin, S., Cui, L., Dong, Y., ... & Tong, Y. (2021). Retrieval of Leaf Area Index by Linking the PROSAIL and Ross-Li BRDF Models Using MODIS BRDF Data. Remote Sensing, 13(23), 4911.
Yang, X., Wang, A., & Jiang, H. (2022). Intelligent Measurement of Frontal Area of Leaves in Wind Tunnel Based on Improved U-Net. Electronics, 11(17), 2730.
Joseph, G. M. D., Mohammadi, M., Sterling, M., Baker, C. J., Gillmeier, S. G., Soper, D., ... & Finnan, J. (2020). Determination of crop dynamic and aerodynamic parameters for lodging prediction. Journal of Wind Engineering and Industrial Aerodynamics, 202, 104169.
Wang, X., Yan, S., Wang, W., Liubing, Y., Li, M., Yu, Z., ... & Hou, F. (2023). Monitoring leaf area index of the sown mixture pasture through UAV multispectral image and texture characteristics. Computers and Electronics in Agriculture, 214, 108333.
Sterling, M., Baker, C., Joseph, G., Gillmeier, S., Mohammadi, M., D Blackburn, G., ... & Sonder, K. (2018, March). Mitigating yield losses due to the lodging of cereal crops. In Proceedings of the International Workshop on Wind-Related Disasters and Mitigation Tohoku University, Sendai, Japan (pp. 11-14).
C. Py, E. De Langre, B. Moulia A frequency lock-in mechanism in the interaction between wind and crop canopiesJ. Fluid Mech., 568 (2006), pp. 425-449
I also encourage the authors to give an outline of the paper at the end of the introduction to let the reader know what to expect.
The methodology employed is notably well-structured. The results and discussion section is commendable for its clarity and depth. The visual aids aid in understanding the outcomes and provide a robust basis for evaluating the effectiveness of the proposed enhancements.
Author Response
Point-to-point responses to the reviewer's comments #4
The topic is interesting and challenging at the same time. I understand this research has approached a very hard task and seems to successfully deliver the aim. The paper is well-written, and the results are interesting. One of the notable strengths of the paper lies in its thorough literature review, which provides a solid foundation for understanding the LAI/models.
Reply: Thank you for your valuable feedback on our manuscript. We appreciate the time and effort you have invested in reviewing our work. We have carefully considered your comments and suggestions, and we would like to address each of them below:
The paper starts well and in the second paragraph, the storyline is well followed. Nevertheless, at the beginning of the third paragraph, I felt almost lost and while narrowing down the topic, the paper did not consider different aspects of the topic well. I believe at the beginning of the literature a bigger picture of measurement of leaf area could be provided before abruptly moving to the models. While looking at the literature I found the following papers might be useful for a wider picture of the topic at the beginning. I hope they help.
Rahimikhoob, H., Delshad, M., & Habibi, R. (2023). Leaf area estimation in lettuce: Comparison of artificial intelligence-based methods with image analysis technique. Measurement, 222, 113636.
Mora, M., Avila, F., Carrasco-Benavides, M., Maldonado, G., Olguín-Cáceres, J., & Fuentes, S. (2016). Automated computation of leaf area index from fruit trees using improved image processing algorithms applied to canopy cover digital photograpies. Computers and Electronics in Agriculture, 123, 195-202.
Zhang, X., Jiao, Z., Zhao, C., Yin, S., Cui, L., Dong, Y., ... & Tong, Y. (2021). Retrieval of Leaf Area Index by Linking the PROSAIL and Ross-Li BRDF Models Using MODIS BRDF Data. Remote Sensing, 13(23), 4911.
Yang, X., Wang, A., & Jiang, H. (2022). Intelligent Measurement of Frontal Area of Leaves in Wind Tunnel Based on Improved U-Net. Electronics, 11(17), 2730.
Joseph, G. M. D., Mohammadi, M., Sterling, M., Baker, C. J., Gillmeier, S. G., Soper, D., ... & Finnan, J. (2020). Determination of crop dynamic and aerodynamic parameters for lodging prediction. Journal of Wind Engineering and Industrial Aerodynamics, 202, 104169.
Wang, X., Yan, S., Wang, W., Liubing, Y., Li, M., Yu, Z., ... & Hou, F. (2023). Monitoring leaf area index of the sown mixture pasture through UAV multispectral image and texture characteristics. Computers and Electronics in Agriculture, 214, 108333.
Sterling, M., Baker, C., Joseph, G., Gillmeier, S., Mohammadi, M., D Blackburn, G., ... & Sonder, K. (2018, March). Mitigating yield losses due to the lodging of cereal crops. In Proceedings of the International Workshop on Wind-Related Disasters and Mitigation Tohoku University, Sendai, Japan (pp. 11-14).
C. Py, E. De Langre, B. Moulia A frequency lock-in mechanism in the interaction between wind and crop canopiesJ. Fluid Mech., 568 (2006), pp. 425-449
Reply: Thank you for your valuable feedback on our manuscript.
We acknowledge your concern regarding the clarity of the third paragraph. We will revise the manuscript to ensure a smoother transition and better coherence between the different aspects of the topic. We reorganized the content to provide a more gradual and logically structured introduction to the measurement of leaf area before delving into the models. We incorporated these references into our literature review to strengthen the context and depth of our discussion.
The second paragraph of the revised manuscript discusses both direct and indirect methods for LAI measurement, as outlined below:
“The measurement of LAI can be achieved through various approaches [16]. Direct methods entail partially or completely defoliating the canopy to assess the total leaf area of plants or trees, which involving destructiveness and a time-consuming process [17,18]. Indirect approaches employ mathematical algorithms to characterize the passage of light through the canopy, utilizing Beer's Law for estimating the total leaf area[16,19]. In recent years, technological advancements have revolutionized the way we measure leaf area. Surface reflectance data captured by moderate-resolution sensors has been employed in the generation of several global LAI products [20-24], and these products have been extensively validated using field measurements and upscaled high-resolution LAI reference maps [25-29]. These technologies offer the potential to enhance our understanding of vegetation dynamics over vast geographical areas, providing valuable information for global climate change studies. However, accurate measurements face challenges posed by factors like cloud cover, satellite orbit constraints, and the requirement for precise multi-angle data[30].”
I also encourage the authors to give an outline of the paper at the end of the introduction to let the reader know what to expect.
Reply:Thank you for your thoughtful feedback. We appreciate your suggestion to provide an outline of the paper at the end of the introduction to guide the reader. We believe this is a valuable addition and will certainly incorporate a concise outline to provide readers with a clear overview of the paper's structure and content.
The last paragraph of the introduction is rewritten as “This study focuses on the collaborative use of the RTLSR_C and PROSAIL models, specifically examining the PROSAIL model's sensitivity to varying LAI at different observation positions. Our emphasis lies in ensuring consistency between the PROSAIL and RTLSR_C models across diverse observation points, with a primary goal of identifying optimal viewing angles for improved inversion results. The paper start with an introduction highlighting the importance of LAI estimation. The methodology section details the coupling of models and PROSAIL sensitivity analysis. Data utilization involves high-quality MODIS BRDF products and an improved look-up table for LAI inversion. Results from optimal viewing angles are presented, followed by validation against ground-based LAI measurements, high-resolution LAI maps, and MODIS LAI products.”
The methodology employed is notably well-structured. The results and discussion section is commendable for its clarity and depth. The visual aids aid in understanding the outcomes and provide a robust basis for evaluating the effectiveness of the proposed enhancements.
Once again, we appreciate your insightful suggestions.

Reviewer 5 Report
Comments and Suggestions for Authors
This work is a useful advance on using satellite imagery to estimate Leaf Area Index, especially with the finding that the results from30 selected viewing angles were comparable in accuracy as previous algorithms based on 397 viewing angles, leading to reduced workload.
Specific comments:
Line 18: “..physically-based Bidirectional Reflectance Distribution Function (BDRF) models..”
Line 114: “There are also studies suggesting that…”
Line 468: use same number of significant figures: “..0.354 to 0.340..”
Line 510-11: “..is that this study searched for the optimal…”
In lines 471 and in 584, it states that there is a high level of consistency when LAI values less than 3.5. Is this a result of reflectance changing substantially as LAI increases from 0.5 to 3.5 or so but reflectance changes very little at LAI values above 4.5 (Figure 3a, 3c)? If so, mention this.
Author Response
Point-to-point responses to the reviewer's comments #5.
This work is a useful advance on using satellite imagery to estimate Leaf Area Index, especially with the finding that the results from30 selected viewing angles were comparable in accuracy as previous algorithms based on 397 viewing angles, leading to reduced workload.
Reply: Thank you very much for your positive comments and precious suggestions that helped to improve the manuscript. We revised our manuscript and responded to the comments as follows.
Specific comments:
Line 18: “..physically-based Bidirectional Reflectance Distribution Function (BDRF) models..”
Reply: We appreciate your suggestion, and we've made adjustments accordingly.
Line 114: “There are also studies suggesting that…”
Reply: We appreciate your suggestion, and we've made adjustments accordingly.
Line 468: use same number of significant figures: “..0.354 to 0.340..”
Reply: We appreciate your suggestion, and we've made adjustments accordingly.
Line 510-11: “..is that this study searched for the optimal…”
Reply: We appreciate your suggestion, and we've made adjustments accordingly.
In lines 471 and in 584, it states that there is a high level of consistency when LAI values less than 3.5. Is this a result of reflectance changing substantially as LAI increases from 0.5 to 3.5 or so but reflectance changes very little at LAI values above 4.5 (Figure 3a, 3c)? If so, mention this.
Reply: Thank you for your suggestions. We have incorporated this potential reason into the results and discussion.
The revised manuscript has undergone the following modifications:
Section 3.1.1 “It shows that as LAI is low, there are significant changes in surface reflectance with the increase in LAI. However, as LAI further increases (>3.5), the reflectance tends to remain relatively constant.”
The second paragraph of section 3.4 “This can be attributed to the use of the look-up table method, the uniform distribution of vegetation parameters employed in the PROSAIL model and the reflectance simulated by the PROSAIL model shows relatively small variations when LAI is greater than 3.5.”
The first paragraph of section 5. “This may be attributed to the PROSAIL model's ability to capture the anisotropic characteristics of the Earth's surface, the algorithm's utilization of uniformly distributed input parameters, and the minimal variations in PROSAIL-simulated reflectance at high LAI.”
